# Loss of Lymphotoxin Alpha-Expressing Memory B Cells Correlates with Metastasis of Human Primary Melanoma

**DOI:** 10.3390/diagnostics11071238

**Published:** 2021-07-12

**Authors:** Franziska Werner, Christine Wagner, Martin Simon, Katharina Glatz, Kirsten D. Mertz, Heinz Läubli, Erika Richtig, Johannes Griss, Stephan N. Wagner

**Affiliations:** 1Laboratory of Molecular Dermato-Oncology and Tumor Immunology, Department of Dermatology, Medical University of Vienna, 1090 Vienna, Austria; franziska.werner@meduniwien.ac.at (F.W.); christine.wagner@meduniwien.ac.at (C.W.); martin.simon@meduniwien.ac.at (M.S.); 2Institute of Medical Genetics and Pathology, University Hospital Basel, University of Basel, 4031 Basel, Switzerland; katharina.glatz@unibas.ch; 3Institute of Pathology, Cantonal Hospital Baselland, 4410 Liestal, Switzerland; kirsten.mertz@ksbl.ch; 4University of Basel, 4001 Basel, Switzerland; 5Laboratory for Cancer Immunotherapy, Department of Biomedicine and Medical Oncology, Department of Internal Medicine, University Hospital Basel, 4031 Basel, Switzerland; heinz.laeubli@unibas.ch; 6Department of Dermatology, Medical University of Graz, 8036 Graz, Austria; erika.richtig@medunigraz.at; 7Department of Dermatology, Medical University of Vienna, 1090 Vienna, Austria; johannes.griss@meduniwien.ac.at

**Keywords:** tumor-associated B cells, memory B cells, activated B cells, human melanoma, tumor microenvironment, lymphotoxin alpha, Interleukin-10, multiplex immunohistochemistry, spatiotemporal dynamics

## Abstract

Activated antigen-experienced B cells play an unexpected complex role in anti-tumor immunity in human melanoma patients. However, correlative studies between B cell infiltration and tumor progression are limited by the lack of distinction between functional B cell subtypes. In this study, we examined a series of 59 primary and metastatic human cutaneous melanoma specimens with B cell infiltration. Using seven-color multiplex immunohistochemistry and automated tissue imaging and analysis, we analyzed the spatiotemporal dynamics of three major antigen-experienced B cell subpopulations expressing lymphotoxin alpha (LTA/TNFSF1) or interleukin-10 (IL-10) outside tertiary lymphoid structures. The expression of both LTA and IL-10 was not restricted to a particular B cell subtype. In primary melanomas, these cells were predominantly found at the invasive tumor-stroma front and, in metastatic melanomas, they were also found in the intratumoral stroma. In primary melanomas, decreased densities of LTA^+^ memory-like and, to a lesser extent, activated B cells were associated with metastasis. Compared with metastatic primary tumors, B cell infiltrates in melanoma metastases were enriched in both LTA^+^ memory-like and LTA^+^ activated B cells, but not in any of the IL-10^+^ B cell subpopulations. Melanoma disease progression shows distinct dynamics of functional B cell subpopulations, with the regulation of LTA^+^ B cell numbers being more significant than IL-10^+^ B cell subpopulations.

## 1. Introduction

There is now conclusive and independent evidence that antigen-experienced activated B cells play an unexpected, essential role in anti-tumor immunity in cancer patients, including the regulation of anti-tumor T cell responses (reviewed in [1]).

Melanomas are malignant primary tumors originating from melanocytes located on the skin or mucosal surfaces. These malignancies are often characterized by high aggressiveness and poor prognosis [2]. In human melanoma, B cells have been shown to regulate CD8^+^ T cell recruitment to tumor sites, promote tumor-associated inflammation and enhance T cell activation by immune checkpoint blockade [3]. Consistently, higher intratumoral pretreatment numbers of B cells and/or tertiary lymphoid structures (TLS) are associated with responses to immune checkpoint blockade in the different therapeutic settings of human melanoma and sarcoma [3,4,5,6,7]. While these data support an immunostimulatory effect, human tumor-associated B cells can also have direct pro-tumorigenic and immunosuppressive activities, e.g., through the secretion of IGF-1, which can induce drug resistance in melanoma cells via FGFR3 activation, and by the suppression of tumor-specific T cell responses through the expression of PD-1, IL-10, or IL-35, as demonstrated in human hepatocellular carcinoma [8,9]. Besides this, phenotyping studies of tissue samples from human melanoma yielded data on both a negative and a positive association between B cell numbers and disease progression and outcome [10,11,12,13,14,15,16]. Taken together, these data imply that there is still a high need to define the spectrum of functions that B cells exert in cancers, including human melanoma.

By secreting cytokines, B cells are able to enhance or suppress a specific immune response. Beyond activation, B cells require specific differentiation as well as additional signals from their surroundings to become relevant cytokine producers. In the case of tumor-associated B cells, cytokine production largely depends on the stimuli they receive from the tumor microenvironment (reviewed in [17]). We have previously shown that melanoma cells can potently induce both differentiation and cytokine secretion in B cells derived from the tumor tissue and peripheral blood of melanoma patients [3]. These results have been further supported at the single-cell level in various B cell subpopulations from human melanoma metastases [8] and in the most recent ex vivo stimulation assays of B cells from melanoma patients [18]. Interestingly, the spectrum of melanoma B cell-derived expression signatures includes cytokines with documented immunostimulatory as well as immunosuppressive activities [3,18]. This observation may help to explain the sometimes contradictory reports of the direct pro- and anti-tumorigenic functions of human B cells and the association of B cell numbers with disease outcome in melanoma patients. Thus, information on the relative contribution and timing of B cell subpopulations with opposing disease-inhibitory and -promoting activities is of paramount importance in understanding the role of B cells in human melanoma. In contrast with other tumors [9,19,20,21,22,23], these data are largely lacking for human melanoma.

Here, we investigated a series of 59 human cutaneous melanoma samples with B cell infiltration to determine the spatiotemporal expression of the two most prominent cytokines with opposing immunoregulatory activities, namely lymphotoxin alpha (LTA/TNFSF1) and Interleukin-10 (IL-10), in different antigen-experienced B cell subpopulations. Using seven-color multiplex immunohistochemistry and automated tissue imaging and analysis of whole tissue sections [3], we identified varying dynamics for LTA^+^ and IL-10^+^ B cell subpopulations during melanoma disease progression.

## 2. Materials and Methods

### 2.1. Patient Cohorts

Whole tissue sections were cut from routine formalin-fixed, paraffin-embedded (FFPE) blocks of cutaneous primary melanomas of Caucasian patients who underwent surgery at the Cantonal Hospital Baselland, Liestal, Switzerland, between 2002 and 2016 and at the Department of Dermatology, Medical University of Graz, Austria, between 2004 and 2020. Tumor samples from Graz were provided by the Biobank Graz of the Medical University. All tumors were obtained with the informed consent of the patients and the pathology files were retrieved with the approval of the local ethics committees (EKNZ vote BASEC 2016-01499 for Liestal; 32-238 ex 19/20 for Graz). Histologic diagnoses were made by board-certified pathologists from the Cantonal Hospital Baselland, Liestal, and board-certified dermatologists at the Department of Dermatology, Medical University of Graz, sometimes together with external board-certified pathologists. Diagnoses were reviewed by two authors of this study, a board-certified pathologist (K.D.M.) and a board-certified dermatologist (S.N.W.). The respective clinicopathologic information is recorded in Table 1 and Table 2. Desmoplastic subtypes of melanoma were not included in this study because they exhibit different clinical behavior [13].

The cohort included 36 patients with primary cutaneous melanoma who were between 31 and 93 years of age at the time of initial diagnosis. Twenty-two patients presented without metastasis within a maximum follow-up of 194 months (mean: 69 months, Table 1). Fourteen patients developed metastasis within a maximum follow-up interval of 167 months (mean: 37 months, Table 2). None of these patients received local or systemic anti-tumor treatment before surgery for the primary tumor.

In addition, we analyzed whole FFPE tissue sections from 23 human melanoma metastases (Appendix A). These samples were collected at the University Hospital Basel between the years 2015 and 2019. These tumor samples were also collected with the informed consent of the patients and the pathology files were obtained with approval from the local ethics committee (EKNZ vote BASEC 2019-00927). Histologic diagnoses were made by board-certified pathologists at the Institute of Pathology, University Hospital Basel, under the direction of K.G. In lymph node metastases, tumor deposits had completely or almost completely replaced lymph node tissue; in the latter, tumor deposits could clearly be histologically separated from the remains of lymphatic tissue.

### 2.2. Seven-Color Multiplex Immunohistochemical Staining

The analysis and readout of tumor tissue were approved by the ethics committee of the Medical University of Vienna (ethics vote 1999/2019). The antibodies used in this study were: anti-CD19 (1:250 dilution, Abcam, Cambridge, UK, rabbit monoclonal IgG, clone EPR5906, catalogue number #134114), CD20 (1:2000, Agilent, Santa Clara, CA, USA, mouse monoclonal IgG2a, clone L26, #M0755), CD38 (1:450, Agilent, mouse monoclonal IgG1, clone AT13/5, #M7077), CD27 (1:500, Abcam, rabbit monoclonal IgG, clone EPR8569, #ab131254), LTA (1:4800, Proteintech, Manchester, UK, rabbit polyclonal IgG, #13111-1-AP) and IL-10 (1:800, Proteintech, rabbit polyclonal IgG, #20850-1-AP). The establishment of staining parameters for these antibodies and seven-color multiplex immunohistochemistry were performed on whole-tissue sections from routine FFPE blocks, as described by us previously [3,24,25]. In brief, each antibody was first established on FFPE tissue sections from human tonsil, the gold standard for lymphocyte antigen detection in pathology. Thereafter, each antibody was assigned to one of the fluorophores Opal 520, Opal 540, Opal 570, Opal 620, Opal 650 and Opal 690 (Akoya Biosciences, Marlborough, MA, USA). In a second round of staining on tonsil tissue, the signals obtained from tyramide signal amplification-based visualization of primary antibodies were matched to the staining signal intensities obtained by conventional immunofluorescence staining by individual dilution of each primary antibody. For multiplex stainings, we performed six iterative rounds of immunohistochemical stainings—one for each antibody—on deparaffinized tissue sections. After 30 min of heat-induced antigen retrieval with either citrate buffer (pH 6.0) or EDTA buffer (pH 9.0), the sections were fixed with 7.5% neutralized formaldehyde (SAV Liquid Production, Flintsbach, Germany), blocked with 20% normal goat serum (Agilent, X0907) and then incubated with the respective primary antibody. Primary antibodies were visualized with corresponding biotinylated anti-mouse or -rabbit secondary antibodies (Agilent, K5003), Streptavidin-HRP (Agilent, K5003) and Opal fluorophore dye (Akoya). After six consecutive rounds of immunostainings, nuclei were counterstained with DAPI (PerkinElmer, Waltham, MA, USA, FP1490) and slides mounted with PermaFluor fluorescence mounting medium (Thermo Fisher Scientific, Waltham, MA, USA).

Negative controls included the use of isotype instead of primary antibodies and staining without primary antibodies. Single antibody stainings were run in parallel to control for false positive (incomplete stripping of antibody-tyramide complexes) and false negative results (antigen masking by multiple antibodies, “umbrella-effect”), as well as for spillover effects (detection of fluorophores in adjacent channels), as previously described [3,24,25]. Reproducibility was controlled by a reference slide in each run, and antibody batches were not changed in this study.

### 2.3. Automated Acquisition and Quantification of LTA^+^ and IL-10^+^ B Cell Subpopulations

We scanned multiplexed slides of whole-tumor sections on a Vectra 3 Automated Quantitative Pathology Imaging System (version 3.0.5., Akoya), which were then analyzed after spectral unmixing with the inForm^®^ Tissue Finder™ (version 2.4.1, Akoya) [3,24,25]. Tumor tissue areas with ulceration or with densely packed clusters of lymphoid cells, e.g., in TLS, did not allow for a clear assignment of fluorophore signals to individual cells and were excluded from our analysis. When present, remnants of lymphatic tissue in nodal metastases were also excluded.

Based on differential expression of CD19, CD20, CD38 and CD27, three B cell phenotypes were determined along the lines we described previously [3]: (i) CD19^+^ CD20^−^ CD38^−^ CD27^+^ as activated B cells, (ii) CD19^+^ CD20^+^ CD38^−^ CD27^var^ as memory-like B cells, (iii) CD19^+^ CD20^−^ CD38^+^ as plasmablast-/ plasma cell-like cells or antibody secreting cells and (iv) other cell types. Based on expression signals for LTA or IL-10, these B cell subpopulations were further classified as LTA^+^ or IL-10^+^ (Figure 1, Appendix A). Cancer tissues including melanoma are known for strong plasma positivity for cytokines/chemokines including IL-10 [26,27]. Thus, immunostaining for secreted cytokines/chemokines often shows staining of the tissue surrounding cells with specific cytoplasmic expression. Therefore—as described above—control staining with isotype-matched antibodies and single antibody stainings were used to control for both false-positive results and spillover effects. In addition, we defined background staining as the weak signal obtained for fibroblasts in the tumor microenvironment. These cells are known to produce very low—if any—levels of LTA and IL-10 themselves [26,27], but can bind cytokines via their corresponding receptors on the cell surface, e.g., anchoring heterotrimeric LTA/B via TNFR3. We, therefore, randomly selected 50 stamps from 10 different tissue samples in which we determined the mean and standard deviation of signal intensities for LTA and IL-10 in a total of 250 intratumoral fibroblasts. Only B cells with signal intensities above the upper standard deviation from the mean of fibroblasts were considered positive. As CD27 expression can be downregulated on tumor-infiltrating B cells [28,29], the number of activated B cells may be underrepresented. All phenotyping and subsequent quantifications were performed blinded to sample identity.

### 2.4. Statistical Evaluation

Cell-level data (i.e., intensities per channel for the cell compartments “membrane”, “cytoplasm”, and “nucleus”) were exported from inForm^®^ Tissue Finder™ (version 2.4.1, Akoya) as text files and processed using R (version 4.0.3). Intensity thresholds were manually adjusted after an inspection of each slide. Based on these thresholds, markers were defined as positive or negative and cells were assigned to specific phenotypes based on the above marker combinations.

## 3. Results

### 3.1. Experimental Strategy

B cells do not homogeneously infiltrate primary human melanomas throughout the tumor tissue, but preferentially at the invasive tumor front, and also sometimes intratumorally, in a scattered and patchy manner [3,12,24]. In melanoma metastases, B cells are predominantly found in the intratumoral stroma around tumor nests and in the peritumoral stroma at the invasive tumor front [3]. We, therefore, performed seven-color multiplex immunohistochemistry and automated tissue imaging and analysis on whole-tissue sections from human melanoma samples.

Our previous work on the spatiotemporal distribution of B cell subpopulations in primary human melanoma further showed the presence of activated and memory-like B cells, as well as of plasmablast-like antibody secreting cells, whereas other subpopulations such as plasma cell-like antibody secreting cells, germinal center-like and transitional/regulatory-like B cells were rarely found outside TLS [3]. We, therefore, decided to focus our analyses on the detection of the three first-mentioned B cell subpopulations by differential expressions of CD19, CD20, CD27 and CD38. In contrast to primary melanomas, melanoma metastases contain also considerable numbers of plasma cells. Here, the aforementioned markers cannot differentiate between plasmablast- and plasma cell-like antibody secreting cells. To allow for an analysis of the different B cell subpopulations within individual tumor samples, we included only tumor samples in which we could detect at least 50 B cells per mm^2^ by CD19 and/or CD20 immunoreactivity (36 primary melanomas, 23 melanoma metastases) in this comparison.

We then determined the densities (cells/mm^2^) of each of the LTA^+^ and IL-10^+^ B cell subpopulations in the primary melanomas and their association with the most important categorical clinicopathologic parameters. Due to the sometimes considerable size of the melanoma metastases, we decided to analyze B cell subtypes there in areas with dense immune cell infiltrates, but not to evaluate B cell numbers per entire tissue area. Therefore, we compared the CD19^+^ and/or CD20^+^ B cell infiltrates of metastases with those of primary tumors in terms of their composition (relative frequencies) for LTA^+^ and IL-10^+^ B cell subpopulations.

### 3.2. Metastasis of Primary Human Melanoma Is Associated with a Decrease in the Number of Intratumoral LTA^+^ Activated and Memory-Like B Cells

The expression of LTA and IL-10 was not restricted to a distinct B cell subtype but detected in each of the analyzed B cell subpopulations, namely activated and memory-like B cells and antibody secreting cells (Figure 1, Appendix A). These cells were predominantly found at the invasive tumor-stroma front of primary human melanomas, and sometimes associated with a patchy intratumoral infiltration. Thus, their distribution followed the previously described infiltration pattern of CD20^+^ B cells and of activated and memory-like B cell subpopulations [12,24].

We detected LTA^+^ B cell populations in all 36 primary melanoma samples (100%, Table 3). The cell densities were highest for LTA^+^ antibody secreting cells, followed by LTA^+^ memory-like and activated B cells. IL-10^+^ B cells were also present in all 36 primary melanoma samples (100%). Again, the densities were highest for IL-10^+^ antibody secreting cells, followed by IL-10^+^ memory-like and activated B cells, but generally lower than those of their LTA^+^ counterparts (Table 3).

Primary tumor samples included both primary tumors that metastasized and those that did not. We compared these two tumor subgroups with each other because metastasis is the most important prognostic factor for melanoma patients.

Primary human melanomas that metastasized had significantly fewer LTA^+^ memory-like B cells (mean 0.612 ± 1.032 vs. 7.419 ± 14.398 cells/mm^2^, *p* = 0.04, CI 95% = 0.1775 to 3.8572, Bonferroni-corrected Wilcoxon Rank Sum test) and some minor trend towards reduced LTA^+^ activated B cells (mean 1.495 ± 2.497 vs. 6.384 ± 11.78, *p* = 0.19, CI 95% = 0.0318 to 5.0355, Bonferroni-corrected Wilcoxon rank sum test) than primary melanomas without metastasis (Figure 2, Table 3). Differences were not found for LTA^+^ and IL-10^+^ antibody secreting cells and IL-10^+^ activated and memory-like B cells.

We have previously shown that the metastasis of primary melanomas is associated with the decrease in memory-like B cell numbers [24]. To test for a preferential decrease in LTA^+^ memory-like B cells in primary tumors with metastasis, we also compared the relative frequencies of LTA^+^ vs. IL-10^+^ cells for different dynamics within the memory-like B cell subpopulation. While the relative frequency of IL-10^+^ cells in primary tumors with metastasis did not change (Bonferroni-corrected *p* = 0.61, CI 95% −0.0119 to 0.0154, Wilcoxon rank sum test), the relative abundance of LTA^+^ cells rather decreased (Bonferroni-corrected *p* = 0.1, CI 95% 0.0 to 0.1667, Wilcoxon rank sum test).

When primary melanoma samples were stratified for another four prognostically important categorical clinicopathologic parameters, we observed a significant association of LTA^+^ memory-like B cell numbers with low Breslow depth (*p* = 0.05, Bonferroni-corrected Wilcoxon Rank Sum test), but not with age, sex and ulceration. Differences in the cell densities of the other LTA^+^ or IL-10^+^ B cell subpopulations were not found (Appendix A).

Thus, LTA and IL-10 are not expressed in a distinct population, but in all analyzed B cell subpopulations and metastasis of primary melanoma is associated with a decrease in LTA^+^ memory-like and, to a minor degree, in LTA^+^ activated B cells but not in IL-10^+^ B cell subpopulations.

### 3.3. Frequencies of LTA^+^ Memory-Like B Cell Subpopulations Change with Melanoma Disease Progression

We have previously shown that the composition of B cell subpopulations also changes with disease progression from primary to metastatic tumor sites [24]. Therefore, we compared primary human melanomas and melanoma metastases for the composition of LTA^+^ and IL-10^+^ B cell subpopulations. As in primary tumors, neither the expression of LTA nor IL-10 was restricted to a distinct B cell subset here.

LTA^+^ and IL-10^+^ B cell subpopulations were mainly found in the intratumoral stromal septa and the peritumoral stroma of melanoma metastases. This is consistent with the distribution previously described for CD20^+^ B cells and of antigen-experienced B cell subpopulations [12,24].

We found LTA^+^ B cell populations in all 23 metastatic melanoma samples (100%). Relative frequencies were highest for LTA^+^ activated B cells, followed by LTA^+^ antibody secreting cells and memory-like B cells. IL-10^+^ B cells were also present in all 23 metastatic melanoma samples (100%). Here, the relative frequencies were highest for IL-10^+^ memory-like B cells, followed by IL-10^+^ antibody secreting cells and activated B cells. As seen in primary tumors, frequencies were much lower than those of their LTA^+^ cell counterparts.

Next, we compared the relative frequencies of LTA^+^ memory-like and activated B cells and antibody secreting cells in melanoma metastases and primary tumors, and only found significant differences in the metastasized primary tumors. Here, the metastases contained higher frequencies of both LTA^+^ memory-like B cells (mean 0.03 ± 0.04 vs. 0.005 ± 0.007, *p* < 0.01 (adjusted), CI 95% −0.0118 to −0.0044, Bonferroni-corrected Wilcoxon Rank Sum test) and LTA^+^ activated B cells (mean 0.052 ± 0.067 vs. 0.012 ± 0.012, *p* = 0.01 (adjusted), CI 95% = −0.0379 to −0.0075, Bonferroni-corrected Wilcoxon rank sum test). No significant differences were observed for the relative frequencies of LTA^+^ and IL-10^+^ antibody secreting cells, as well as IL-10^+^ activated and memory-like B cells (Figure 3, Appendix A).

We have previously shown that both TLS density and maturation as well as the composition of B cell subpopulations vary between different metastatic tumor sites, particularly between lymph node and skin [24,25]. When we compared the relative frequencies of LTA^+^ and IL-10^+^ B cell subpopulations in metastatic lymph nodes and skin samples, we found enrichment of IL-10^+^ activated B cells at lymph node sites because we could not detect IL-10^+^ activated B cells at metastatic skin sites (mean 0.005 ± 0.005 vs. 0 ± 0 cells, *p* = 0.04, CI 95% 3 × 10^−4^ to 0.0113, Bonferroni-corrected Wilcoxon Rank Sum test). All other B cell subpopulations showed no significant differences (Figure 4).

## 4. Discussion

Using seven-color multiplex immunohistochemistry and an automated tissue imaging and analysis approach, we show (i) that the expression of LTA and IL-10 is not limited to a particular B cell subtype, but can be detected in any of the analyzed antigen-experienced B cell subpopulations, (ii) that metastasis of primary tumors is associated with a decrease in the densities of LTA^+^ memory-like B cells and, to a minor degree, LTA^+^ activated B cells, but not IL-10^+^ B cell subpopulations, (iii) that metastatic sites are enriched for LTA^+^ memory-like and LTA^+^ activated B cells compared with metastatic primary tumor sites, (iv) that the B cell infiltrates in metastases at lymph node and skin sites have comparable compositions for LTA^+^ and IL-10^+^ B cell subpopulations, and (v) that the composition of B cell infiltrates for IL-10^+^ B cell subpopulations does not change significantly with disease progression.

To date, in human melanoma, correlative studies between B cell infiltration and tumor progression are limited by the fact that they do not differentiate between functional B cell subtypes. We have shown that human melanoma cells can induce both B cell receptor and CD40 signaling in activated B cells [3]. Among other mechanisms [30], the balance between B cell receptor and CD40 signaling in activated B cells critically determines the production of immunostimulatory vs. immunoinhibitory cytokines such as LTA vs. IL-10 (reviewed in [17]).

LTA is secreted as a bioactive homotrimer (LTA3) that, like TNFA, binds to TNFR1, TNFR2, HVEM/TNFRSF14 and, unlike TNFA, as a membrane-anchored hetero-trimer with LTB (LTA1B2, LTA2B1) that binds to LTβ-R/TNFR3. Both TNFR1 and TNFR2 are potent pro-inflammatory, pro-apoptotic and, when caspases are inhibited, necrotic-like cell death-inducing receptors, but TNFR1 in particular seems to be responsible for LTA-mediated cytotoxicity in vitro (reviewed in [31]). In transgenic murine models, LTA showed a pro-inflammatory function, e.g., through the induction of cell adhesion molecules and chemokines in endothelial cells (reviewed in [32,33]). In syngeneic transplantation models with melanoma cells, LTA supported natural killer (NK) cell-dependent anti-tumor activity [34] and, as a recombinant GD2 scFv-LTA fusion protein, stimulated adaptive T cell responses with clonal expansion of different melanoma-reactive T cell receptors in newly induced TLS [35]. In line with these data, B cell-expressed LTA1B2 controlled the organization of ectopic TLS at homeostasis and, following immune challenge, the activation of TLS for the generation of both adaptive T helper (T_H_) 1 and T_H_2 immune responses (reviewed in [36]).

We here demonstrate that several B cell subpopulations are a source of LTA in human melanoma and the decreased density of LTA^+^ B cells is associated with primary tumor metastasis. Whether the expression of LTA in B cells promotes tumor cell cytotoxicity, inflammation and induction of functional TLS in human melanoma remains to be established; our initial data, however, point to the presence mainly of early immature but not fully mature TLS in primary human melanomas [25]. Additionally, the association of a decreased density of LTA^+^ B cells with primary tumor metastasis needs to be evaluated for other human cancer types, particularly in light of the reported promotion of androgen-independent tumor growth in a murine prostate cancer model through B cell-derived LTA/B, which activated pro-tumorigenic IKK-alpha and STAT3 signaling in cancer cells [37].

Immunosuppressive IL-10^+^ B cells may accumulate at inflammatory or tumor sites through increased cell survival via the enhanced expression of transcription factor C-MAF upon restraint of SLAMF5 receptor-mediated signaling [38]. Through the secretion of IL-10, these B cells can attenuate T_H_1, T_H_2 or T_H_17 cell-mediated immune responses in murine models of autoimmune, inflammatory and infectious diseases (reviewed in [36]) and inhibit anti-tumor cytotoxic T cell responses in mouse models of colorectal cancer and melanoma (reviewed in [39,40]). Reported mechanisms include the suppression of IFN-γ and IL-17 production in T_H_1, NK and T_H_17 cells, respectively; interference with differentiation of T cells into T_H_17 cells and proliferation of CD4^+^ T cells; induction of regulatory T cells and IL-10^+^ suppressive T cells from effector T cells; suppression of monocyte proliferation and IL-12 production in dendritic cells (reviewed in [40,41,42]).

In parallel with studies in mouse models, IL-10^+^ B cell numbers have also been associated with human cancer progression. Examples are IL-10^+^ B cells in tumor tissues of gastric cancer [20], squamous cell carcinoma of the tongue [21] and ascites from ovarian cancer [22]; IgA^+^ CD138^+^ PD-L1^+^ IL-10^+^ B cells in prostate cancer [23]; CD5^high^ CD24^−/+^ CD27^high/+^ CD38^dim^ PD-1^high^ B cells producing IL-10 in human hepatocellular carcinoma [9]; Granzyme B^+^ CD38^+^ CD1d ^+^ IgM ^+^ CD147^+^ B cells expressing IL-10 and IDO in breast, cervical and ovarian carcinomas [19]. In contrast to these studies, our results now only show low numbers of IL-10^+^ B cells in human melanoma, numbers which were considerably lower than those of LTA^+^ B cells. An association of IL-10 expression with a specific B cell subpopulation was not found, nor was an association of melanoma metastasis with an increased number of IL-10^+^ B cells. Given the poorly recognized but still substantial body of data on pro-inflammatory functions of IL-10 and other IL-10 cytokine family members (reviewed in [43,44]), our data support further evaluation of B cell-derived IL-10 in human melanoma.

## 5. Conclusions

Taken together, this study expands the knowledge of the spectrum of functional B cell subpopulations and their spatiotemporal dynamics in human melanoma. Interestingly, our data indicate that melanoma progression appears to be associated with changes in the number of LTA^+^ rather than IL-10^+^ B cell numbers, suggesting a more relevant functional role for LTA^+^ B cells. Moreover, the expression of immunoregulatory cytokines in multiple, rather than just one, B cell subpopulation raises the interesting question of the extent to which one B cell subpopulation could functionally replace the others during melanoma progression. The different dynamics of putative immunostimulatory and immunosuppressive B cell subpopulations during melanoma progression underscores the importance of this type of analysis for other cancer types.

## Figures and Tables

**Figure 1 diagnostics-11-01238-f001:**
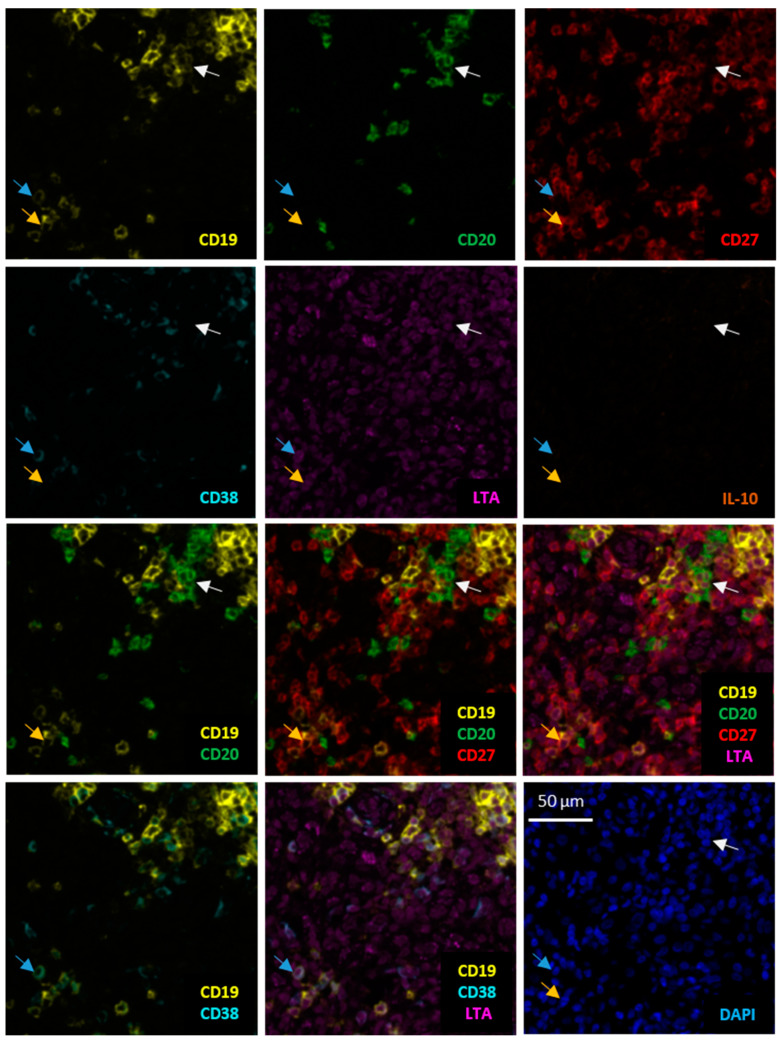
LTA^+^ B cell subpopulations in human melanoma. Identification of a LTA^+^ CD19^+^ CD20^+^ CD27^−^ CD38^−^ memory-like B cell (white arrow), a LTA^+^ CD19^+^ CD20^−^ CD27^+^ CD38^−^ activated B cell (yellow arrow) and a LTA^+^ CD19^+^ CD20^−^ CD27^−^ CD38^+^ antibody secreting cell (blue arrow). Serial images of the same cells for the different cell markers and the cytokines LTA and IL-10 in the two upper rows; composite images for marker combinations are given in the two lower rows, together with respective LTA staining and negative staining for IL-10. DAPI nuclear staining in the lower right. Arrows depict the same cell, being representative of the respective B cell subpopulation. Scale bar represents 50 µm.

**Figure 2 diagnostics-11-01238-f002:**
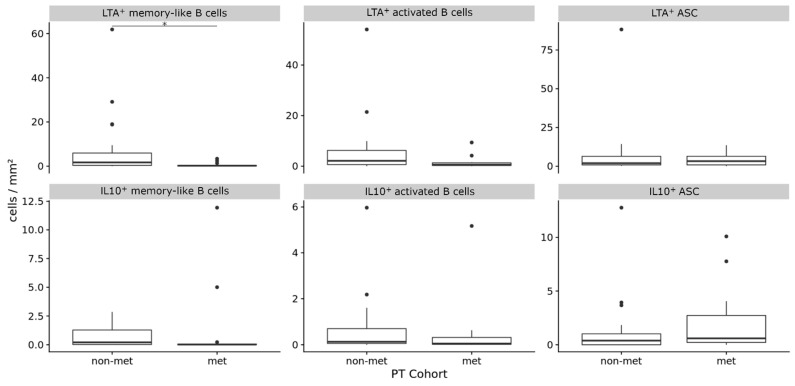
The frequencies (cells/mm^2^) of LTA^+^ and IL-10^+^ B cell subpopulations in primary human melanomas and their association with metastasis. Box plots comparing primary tumors that did not metastasize (non-met) versus those that metastasized (met). In boxplots, lower and upper hinges correspond to the first and third quartiles and center lines to medians. Lower and upper whiskers extend from the hinge to the largest value within 1.5 times the interquartile range. Outliers are shown as black circles, * *p* ≤ 0.05. ASC = antibody secreting cells.

**Figure 3 diagnostics-11-01238-f003:**
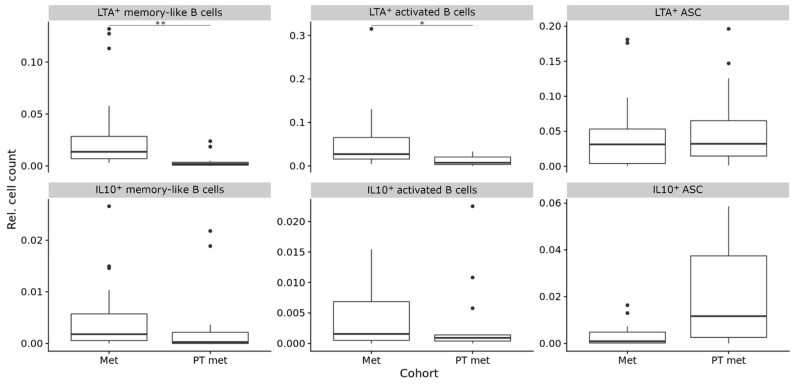
Composition (relative frequencies) of LTA^+^ and IL-10^+^ B cell subpopulations in metastases compared to primary tumors. Here, plots are shown for comparison to metastasized primary tumors, the full dataset, including comparison to non-metastasized primary tumors, is available in Appendix A. In boxplots, lower and upper hinges correspond to the first and third quartiles and center lines to medians. Lower and upper whiskers extend from the hinge to the largest value within 1.5 times the interquartile range. Outliers are shown as black circles, * *p* ≤ 0.05, ** *p* < 0.01. ASC = antibody secreting cells.

**Figure 4 diagnostics-11-01238-f004:**
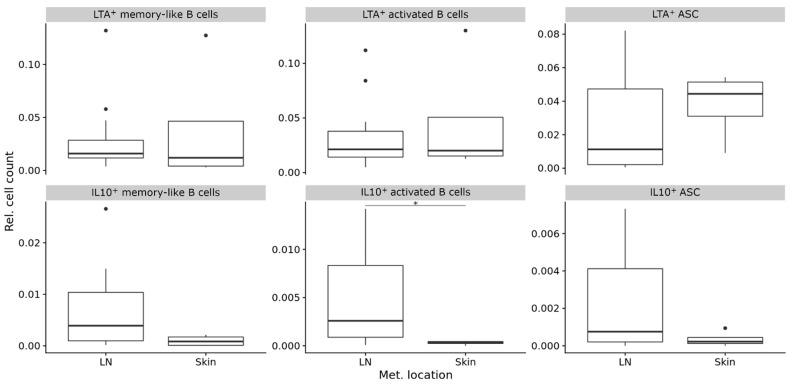
Composition (relative frequencies) of LTA^+^ and IL-10^+^ B cell subpopulations in metastatic tumor sites. Boxplots comparing metastatic lymph node (LN) and skin sites. In boxplots, lower and upper hinges correspond to the first and third quartiles and center lines to medians. Lower and upper whiskers extend from the hinge to the largest value within 1.5 times the interquartile range. Outliers are shown as black circles. * *p* ≤ 0.05. ASC = antibody secreting cells.

**Table 1 diagnostics-11-01238-t001:** Clinicopathologic summary of melanoma patients with primary tumors without subsequent metastasis.

No. of Patients		22
Follow-up (months)	Mean	69
	Median	62
	Range	8–194
Age (years)	Median	69
	Mean	72
	Range	31–93
Breslow depth (thickness in mm)	MeanMedian	3.64 3.38
	Range	0.36–10
Location	Extremities	6
	Head/Neck	0
	Trunk	16
Ulceration	Present	12
	Absent	10
Histotype *	SSM	13
	NM	7
	ALM	1
	NOS	1
Sex	Male	14
	Female	8

* SSM (superficial spreading melanoma), NM (nodular melanoma), ALM (acral lentiginous melanoma), NOS (not otherwise specified).

**Table 2 diagnostics-11-01238-t002:** Clinicopathologic summary of melanoma patients with primary tumors with metastasis.

No. of Patients		14
Follow-up (months) *	Mean	37
	Median	22
	Range	0–167
Age (years)	Median	69
	Mean	71
	Range	38–91
Breslow depth (thickness in mm) **	MeanMedian	9.19 10.40
	Range	1.0–17.0
Location **	Extremities	4
	Head/Neck	1
	Trunk	8
Ulceration **	Present	9
	Absent	4
Histotype ***	SSM	3
	NM	11
Sex	Male	9
	Female	5

* Three samples without follow-up information (after metastasis). ** Three samples without exact information about Breslow depth, one about location, one about ulceration. *** SSM (superficial spreading melanoma), NM (nodular melanoma).

**Table 3 diagnostics-11-01238-t003:** Summary of multiplex immunohistochemistry staining results in primary tumor samples.

	Primary Melanomas without Metastasis	Primary Melanomas with Metastasis
No. of samples	22	14
No. of samples with LTA^+^ B cell subpopulations	22 (100%)	14 (100%)
No. of samples with IL-10^+^ B cell subpopulations	22 (100%)	14 (100%)
No. of LTA^+^ cells/mm^2^ tumor area		
LTA^+^ activated B cells, range:	0–54.069	0–9.368
mean ± sd:	6.384 ± 11.78	1.495 ± 2.497
LTA^+^ memory-like B cells *, range:	0–61.851	0–3.355
mean ± sd:	7.419 ± 14.398	0.612 ± 1.032
LTA^+^ antibody secreting cells, range:	0.251–88.246	0.05–13.568
mean ± sd:	7.538 ± 18.46	4.646 ± 4.541
No. of IL-10^+^ cells/mm^2^ tumor area		
IL-10^+^ activated B cells, range:	0–5.97	0–5.171
mean ± sd:	0.673 ± 1.326	0.518 ± 1.357
IL-10^+^ memory-like B cells, range:	0–2.855	0–11.94
mean ± sd:	0.693 ± 0.914	1.242 ± 3.352
IL-10^+^ antibody secreting cells, range:	0–12.775	0–10.089
mean ± sd:	1.279 ± 2.793	2.167 ± 3.145

* *p* ≤ 0.05 between primary melanomas without vs. with metastasis.

## Data Availability

The data presented in this study are available in this article and the Appendix A.

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
