# Peer review of "Loss of Lymphotoxin Alpha-Expressing Memory B Cells Correlates with Metastasis of Human Primary Melanoma"

_diagnostics, 2021, doi:10.3390/diagnostics11071238_

Round 1
Reviewer 1 Report
The manuscript entitled; “Loss of Lymphotoxin Alpha-Expressing Memory B Cells Correlates with Metastasis of Human Primary Melanoma” is interesting. The authors examined a series of 59 primary and metastatic human cutaneous melanoma specimens for antigen-experienced B cell subpopulations expressing LTA/TNFSF1 or IL-10. The study design and data analysis are appropriate. The data demonstrated that decreased densities of LTA+ memory like B cells were associated with metastasis in primary human melanomas. Overall, distinct dynamics of enriched LTA+ memory-like and LTA+ activated B cells were found to be associated with disease progression compared to IL-10+ B cell subpopulations. I have a few concerns/comments.
- At places, loss or decreased densities of LTA+ B cells have been used interchangeably, which makes it very confusing. Please be consistent.
- While non-significant, there were decreased densities of LTA+ activated B cells in metastatic primary human melanomas. Not sure why this was excluded in the abstract?
Author Response
1. "At places, loss or decreased densities of LTA+ B cells have been used interchangeably, which makes it very confusing. Please be consistent."
The authors thank the reviewer for this advice. The term “loss” was used to describe the state of no longer having as much of something as before. We understand that this may perhaps sometimes confusing and have now removed “loss” from the manuscript except the title.
2. "While non-significant, there were decreased densities of LTA+ activated B cells in metastatic primary human melanomas. Not sure why this was excluded in the abstract?"
Following the recommendation of the reviewer, we have now included the information on activated B cells in the abstract.
Reviewer 2 Report
An interesting original article documenting a series of 59 human cutaneous melanoma samples with B cell infiltration to determine the spatiotemporal expression of lymphotoxin alpha 78 (LTA/TNFSF1) and Interleukin-10 (IL-10).
I have some queries:
Please specify the maker and location of the equipment and statistical programs you used.
Page 1 line 43 you should add: "Melanomas are malignant primary tumors originating from melanocytes located on the skin or mucosal surfaces. These malignancies may be characterized by high aggressiveness and poor prognosis" and cite an article such as: https://doi.org/10.3390/medicina57040359
page 4 line 407-410 should be deleted "Authors should discuss the results and how they can be interpreted from the perspective of previous studies and of the working hypotheses. The findings and their impli409 cations should be discussed in the broadest context possible. Future research directions may also be highlighted."
I would probably split the discussion and create a conclusion paragraph, better highlighting future prospective following this study.
Author Response
“Please specify the maker and location of the equipment and statistical programs you used.”
As far as the authors understand this information was missing for the software “inForm Tissue Finder”. Following the recommendation of the reviewer, we have added this information now. The location of the provider was given once at first appearance in the manuscript.
“Page 1 line 43 you should add: "Melanomas are malignant primary tumors originating from melanocytes located on the skin or mucosal surfaces. These malignancies may be characterized by high aggressiveness and poor prognosis" and cite an article such as: https://doi.org/10.3390/medicina57040359”
Following the recommendation of the reviewer, we have added this information. However, we cited a slightly different publication which seemed more appropriate to us.
“page 4 line 407-410 should be deleted "Authors should discuss the results and how they can be interpreted from the perspective of previous studies and of the working hypotheses. The findings and their impli409 cations should be discussed in the broadest context possible. Future research directions may also be highlighted."
Following the recommendation of the reviewer, we have deleted this paragraph.
“I would probably split the discussion and create a conclusion paragraph, better highlighting future prospective following this study.”
Following the recommendation of the reviewer, we have now added a conclusion section after the discussion.
Round 2
Reviewer 2 Report
The authors responded to all queries. The paper is publishable